# Electrocardiographic Findings in Bitches Affected by Closed Cervix Pyometra

**DOI:** 10.3390/vetsci7040183

**Published:** 2020-11-20

**Authors:** Michela Pugliese, Rocky La Maestra, Annamaria Passantino, Santo Cristarella, Massimo De Majo, Vito Biondi, Marco Quartuccio

**Affiliations:** Department of Veterinary Sciences, University of Messina, 98168 Messina, Italy; mpugliese@unime.it (M.P.); rocky.lamaestra@unime.it (R.L.M.); santo.cristarella@unime.it (S.C.); mdemajo@unime.it (M.D.M.); vito.biondi@unime.it (V.B.); marco.quartuccio@unime.it (M.Q.)

**Keywords:** electrocardiogram, bitch, pyometra, myocardial injury

## Abstract

Pyometra is considered the most common disease in intact bitches, being associated with potentially life-threatening disorders. Myocardial damage is a potentially life-threatening consequence of pyometra. The aim of this study was to describe the electrocardiographic patterns in bitches affected by closed cervix pyometra, to assess the clinical relevance of electrocardiographic changes with the occurrence of pyometra, and to relate their severity with laboratory and clinical findings. A total of 39 bitches with closed cervix pyometra and 10 healthy female dogs were included in this study. During the hospitalization, bitches underwent a complete physical examination. An electrocardiographic examination before the ovariohysterectomy was performed. Blood samples for biochemical and hematological analysis were also evaluated. Bitches suffering pyometra at least one arrhythmia 31/39 (79.4%), sinus tachycardia (22/39, 56.4%), ventricular premature complexes (9/39, 23%), increased amplitude of T wave (7/39, 17.9%), ST depression (4/39, 10.2%), second-degree atrioventricular block (2/39, 5.1%), increase of QT interval (2/39, 5.1%), sinus bradycardia (2/39, 5.1%), and first-degree atrioventricular block (1/39, 2.5%). Some bitches were also detected with low wave amplitude (17/39, 43.5%). Cardiac arrhythmias associated with canine pyometra are frequent events. These data suggest that arrhythmias may be the consequence of one or more factors that can occur during pyometra, such as myocardial damage, electrolyte/metabolic disorders, and/or sepsis.

## 1. Introduction

Canine pyometra is a disorder of the reproductive tract common in countries where elective spaying is not routinely performed [1,2]. The disease affects an average of 19% of non-spayed sexually mature bitches over 8 years of age [1,2]. The risk to develop pyometra is strongly related to genetic factors. In certain breeds, a high incidence of this disease is reported [3,4]. The disease can be classified as open-cervix or closed-cervix pyometra (CCP), with the latter being more severe [1,2,5,6]. The causes of pyometra are complex, and a not yet fully understood hormonal component plays an important role [1,2,7,8]. The administration of estrogen and progestogen seems to make the uterus more vulnerable to infection by opportunistic bacteria, such as *Escherichia coli* (*E. coli*) [7,8,9,10,11]. A healthy uterus normally manages to defend itself effectively from bacteria that have settled when the cervix was open during the estrous phase [1,2,7]. Hormonal alterations in the diestrus or previous uterine pathologies such as endometrial cystic hyperplasia may decrease the uterine clearance of bacteria [1,2,7,12]. Myocardial damage is considered a potentially life-threatening outcome in the course of pyometra due to bacteremia, septicemia, disseminated bacterial infection, endotoxemia, and multiple organ dysfunction [1,13]. Ovariohysterectomy is often the only effective treatment for pyometra, although it may cause worse myocardial cell damage in animals with systemic inflammation and impaired circulation [14,15]. An increase of plasma levels of cardiac troponin, suggestive of myocyte injury, has been reported in bitches affected by pyometra [13,15].

The electrocardiogram (ECG) is considered an essential instrument to determine the origin and frequency of impulse and to evaluate cardiac conduction disorders. This tool is an essential part of pre-surgery check-up, especially in elderly animals, in order to minimize the risks of surgical complications and to reduce mortality and surgical morbidity [16]. In addition, the ECG provides information changes in electrolytes (calcium and potassium) and myocardial oxygenation status, which may be apparent in the course of canine pyometra.

The purpose of this study was to describe electrocardiographic features in bitches affected with CCP and to assess the clinical relevance of ECG changes, as well as to relate the severity of ECG with laboratory findings.

## 2. Materials and Methods

### 2.1. Ethics Statement

The protocol of this study was performed following the standards recommended by the Guide for the Care and Use of Laboratory Animals and Directive 2010/63/EU. The protocol and procedures employed were ethically reviewed and approved by the Ethical Committee of the Department of Veterinary Sciences of the Messina University (approval no. 10/2018).

Informed written consent was obtained from all owners of the bitches at the time they were enrolled at the Veterinary Teaching Hospital (University of Messina).

### 2.2. Study Group

This study enrolled 39 bitches (CCP group) of different breeds (15 crossbreeds, 5 Yorkshire Terriers, 3 Labrador Retrievers, 3 German Shepherds, 2 pit bulls, 2 pugs, 2 poodles, 2 Jack Russell Terriers, 1 Golden Retriever, 1 Shih Tzus, 1 Scotch Collie, 1 Cocker Spaniel, 1 beagle) with CCP diagnosis. The mean age and body weight were 8.4 years (range 4–13 years) and 14.2 kg (range 3.2–29 kg), respectively. Of the CCP group, 28/39 bitches had not had previous pregnancies and 23/39 delivered healthy puppies at term.

The control group (CTR group) included 10 healthy non-spayed adult bitches of different breeds (4 crossbreeds, 2 Yorkshire Terriers, 1 Labrador Retriever, 1 German Shepherd, 1 Jack Russell Terrier, 1 pug). The mean and range for age and body weight were 6.1 years (range 2–9 years) and 15.6 kg (range 3.8–28 kg), respectively. All bitches had never been affected by reproductive or cardiac diseases and had to be spayed.

### 2.3. Procedures

The suspect of pyometra was performed on the basis of medical history, clinical signs, laboratory findings, and abdominal ultrasound examination. The diagnosis was confirmed by the histopathological examination. Blood samples and ECG tracks were detected after the physical examination. In order to better characterize possible heart disease, we placed all dogs under echocardiographic examination.

### 2.4. Clinical Examination

Anamnestic information was collected for all bitches upon arrival at the hospital, followed by physical examination, which included measurement of heart and respiratory rate, cardiac and thoracic auscultation, pain response to abdominal palpation, determination of hydration state, temperature levels, lymph node palpation, assessment of mucosae color, and capillary fill. Bitches fulfilling two or more of the following criteria were defined as presenting as systemic inflammatory response syndrome (SIRS)-positive [17]: hypothermia/hyperthermia (≤37.8 °C or ≥39.7 °C), tachycardia (≥160 bpm), tachypnea (≥40 breaths/min), and leukocytosis/leukopenia (white blood cells (WBC) ≥ 12,000 WBCs/μL or WBC ≤ 4000 WBCs/μL or >10% band neutrophils).

### 2.5. Blood Samples

Blood samples obtained by jugular or cephalic venipuncture from each dog using the vacutainer blood collection system were drawn into 2 different types of tubes: with no additive and containing tripotassium ethylenediaminetetraacetic acid (K_3_EDTA). After collection, samples without additive were centrifuged (ALC 4235 A, Milan, Italy) at 3000 rpm for 20 min. Serum samples obtained after centrifugation were divided into aliquots and processed. Hematological evaluations were performed using a hematology analyzer on the basis of focused flow impedance and flow cytometry (ProCyte Dx, Idexx Laboratories, Westbrook, ME, USA). Serum obtained from centrifugation was used to perform a biochemical examination using a chemistry analyzer (Catalyst Dx Analyzer, Idexx Laboratories, Westbrook, ME, USA). Amino-transferase (ALT), alkaline phosphatase (ALP), glucose (GLU), urea (URE), total protein (TP), albumin (ALB), creatinine (CREA), sodium (Na^+^), potassium (K^+^), and total Ca^2+^ were assayed.

### 2.6. Electrocardiographic Examination

All cardiac tracings were obtained in a quiet place by using a 12-channel ECG recorder (Delta Tre Plus, Cardioline, Italy), with the bitches placed in right lateral recumbency [18]. Electrodes were connected by metallic alligator clips attached directly to the skin at the level of the olecranon on the caudal aspect of the forelimb, and over the patellar ligaments on the cranial aspect of the hindlimbs. Isopropyl alcohol was applied to ensure an adequate contact between electrodes and the skin. Standard bipolar (I, II, and III) and augmented unipolar limb (aVR, aVL, and aVF) leads were recorded using 25 and 50 mm/s paper speed and 1 cm = 1 mV for 5 min. The nomenclature and ECG interpretation were performed according to the standard methods [18]. The mean duration and amplitude of P, QRS, and T waves were determined for all leads using a metal caliper system directly on the thermal paper, considering an average measurement from 5 consecutive complexes for each dog. The QRS duration was measured from its onset to the ST segment. The PR interval was measured from the beginning of the P wave to the beginning of the QRS complex. The QT interval was measured from the beginning of the QRS complex to the end of the T wave.

The sinus rhythm for 20 beats on lead II was manually determined. The mean electrical axis (MEA) in the frontal plane was determined in II and III leads using the Bailey hex-axial method [17]. Sinus arrhythmia and wandering sinoatrial pacemaker were considered normal rhythm for dogs. The arrhythmias were classified according to origin and duration. All recorded ECG values were compared to reference ranges for canine species [18,19].

### 2.7. Histopathological Examination

The diagnosis of pyometra was confirmed by gross and histopathological examination of hematoxylin–eosin-stained sections of formaldehyde-fixated uteri and ovaries. Samples for histopathological examination were obtained from different sites in the uterine horns and body. The histopathological diagnosis referable to pyometra was purulent metritis (acute or chronic) and endometritis.

### 2.8. Statistical Analysis

Statistical analysis was performed using the Statistical Package for the Social Sciences (SPSS) software (Version 17.0, SPSS, Inc., Chicago, IL, USA). Laboratory and electrocardiographic variables were analyzed as quantitative variables and described as mean ± standard deviation (SD). Normality was assessed through the Shapiro–Wilk test. The Mann–Whitney *U*-test for non-parametric data was used to detect statistical differences between the CTR and CCP group for data not presenting a normal distribution. Statistical significance was defined as *p* ≤ 0.05 for all statistical tests used in the study.

## 3. Results

### 3.1. Clinical Variables

Medical history and physical examination for CCP bitches are summarized in Figure 1.

Histological examination of the uterus and ovaries confirmed that all bitches were in diestrus. The cardiac auscultation revealed the presence of murmur in 18/39 (46.1%) bitches. In total, 32/39 (82%) bitches had SIRS. Two SIRS criteria were fulfilled by 24/39 (61.5%), three criteria by 6/39 (15.3%), and all four criteria by 2/39 (5.1%). The murmurs located at the mitral or tricuspidal valve focus ranged between second and third degree. All CTR bitches were healthy and normothermic, with normal general condition, normal hydration status, no pain at the abdominal palpation, and capillary refill time <2 s.

### 3.2. Hematological Variables

Data of hematological variables are summarized in Table 1. CCP bitches had lower red blood cells counts (*p* = 0.032), hemoglobin (*p* = 0.012), packed cell volume (PCV) (*p* = 0.038), mean corpuscular hemoglobin concentration (MCHC) (*p* = 0.041), and platelets (*p* = 0.026) than the CTR group. Higher white blood cell count (*p* = 0.19), neutrophils (*p* = 0.047), and monocytes (*p* = 0.018) were detected in CCP bitches compared to CTR. All hematological variables evaluated in the CTR group were within reference ranges.

### 3.3. Biochemical Variables

Biochemical parameters are shown in Table 2.

Serum albumin in CPP bitches was lower (*p* = 0.05) than the control. Regarding hepatic and renal function serum levels, the ALT (*p* = 0.038), ALP (*p* = 0.041), urea (*p* = 0.046), and creatinine (*p* = 0.028) were higher in CCP than CTR. CCP bitches had a lower serum sodium level (*p* = 0.05) and higher potassium (*p* = 0.05) than CTR. Variables in the CTR group were within the reference ranges.

### 3.4. Electrocardiographic Findings

Quantitative ECG data are summarized in Table 3. All mean values were within reference ranges.

The prevalence of ECG abnormalities was particularly high in bitches affected by CCP, as shown in Table 4.

The variables detected in CTR bitches were within reference ranges. The majority of bitches in the CCP group (31/39, 79.4%) showed ECG abnormalities. Sinus tachycardia was presented in 22/39 (56.4%) bitches; of these, 16/22 (72.7%) had fever. Ventricular premature complexes (VPCs) were observed in 9/39 (23%), atrial fibrillation in 2/39 (5.1%), second-degree atrioventricular block in 2/39 (5.1%), and first degree in 1/39 (2.5%) CCP. VPCs occurred as isolated ectopic events and pairs in all bitches affected. Low-wave amplitude was presented in 17/39 (43.5%) bitches. Increase amplitude of T wave was observed in 7/39 (17.9%) bitches, but in 5/7, there was hyperkalemia (>4.1 mmol/L). ST depression was detected in 4/39 (10.2%). Sinus bradycardia was reported in 2/39 (5.1%). The increase of QT interval (>50% of previous R-R interval) was detected in 2/39 (5.1%) bitches without tachycardia.

## 4. Discussion

CCP is considered a medical emergency because severe subsequent complications such as sepsis, septic shock, peritonitis, disseminated bacterial infection, and multi-organ dysfunction may be related to a fatal outcome [1,2,6]. Mortality of approximately 10% occurs in bitches affected with pyometra, in which the ovariohysterectomy was not performed [1]. Ovariohysterectomy is the treatment of choice in order to prevent recurrences. An appropriate and fast stabilization of the bitch before ovariohysterectomy may ensure a greatly decrease in morbidity and mortality. Myocardial injury is suspected to be a contributing factor to unexpected deaths [13].

Polyuria, polydipsia, palpable enlarge uterus, abdominal pain, lethargy/depression, inappetence/anorexia, and tachycardia are the most common clinical findings recorded in bitches affected with CCP. The hypochromic anemia reported in CCP bitches can be explained by suppressive effects from pyometra on the bone marrow associated with iron deficiency due to continuous blood loss in the uterine lumen [22,23]. Leukocytosis with neutrophilia and an increased number of immature forms are common hematological findings reported in bitches with pyometra [21,22]. Leukocytosis is related to the severity of inflammation and the suppurative nature of the disease. An increase in total leucocyte count and a decrease in lymphocyte count is directly proportional to the severity of the disease in bitches with pyometra [22,23]. Moreover, these findings are frequently related to changes in body temperature and heart rate, suggesting the presence of systemic inflammatory response syndrome [22,23,24]. In humans, several arrhythmias and changes in ECG are described in the course of SIRS. It has been suggested that on basis of arrhythmia onset, there is an alteration of triggered activity related to release of pro-arrhythmic factor during the SIRS, including high catecholamine levels [25]. Hypoalbuminemia associated with hyperglobulinemia occurs, in response to an acute inflammatory reaction [22,23]. A slightly high value of ALT in the CCP group in comparison with CTR suggested hepatic parenchymal damage. Elevated levels of ALP have been considered to be due to intrahepatic cholestasis. Renal dysfunction is described in the CCP bitches as a consequence of endotoxemia, glomerular dysfunction, renal tubular damage, and decreased response to antidiuretic hormone contribution [1,22,23].

Several studies in humans have shown an association between sepsis and various cardiac arrhythmias, focusing on a major incidence of supraventricular arrhythmias [26,27]. The mechanisms involved in the incidence of cardiac arrhythmias in sepsis have not been completely understood, but could include several pathophysiologic pathways impacting the arrhythmogenic substrate, the trigger factors, and the modulation factors. Data of the present study indicate that the arrhythmias and abnormalities of ECG variables commonly occur in canine with closed cervix pyometra. Previous studies evaluating the presence of myocardial injury in bitches with pyometra by troponin assay [13] demonstrated a strong correlation between the increase of cardiac troponin I (CTnI) levels and poor outcomes. Heart muscle injury may occur in bitches affected by pyometra following bacteremia, septicemia, disseminated bacterial infection, and endotoxemia [13,24]. Myocarditis is considered a common complication [1,2,13]. In humans, the presence of myocardial injury induces a reduction of oxygen supply associated with myocardial oxygen consumption, an increase of myocardial wall stress, hypertension, tachycardia, and release of myocardial toxins as a consequence of sepsis [28]. It is possible that the same types of disorders may be present in dogs, identifying the bitches affected by closed cervix pyometra at risk for cardiac events [13,26,28].

Sinus tachycardia was the most common ECG finding in bitches with CCP. Sinus tachycardia can have two different mechanisms: enhanced normal automaticity of sinus pacemaker cells or a re-entry circuit. One or more factors combined can cause sinus tachycardia in dogs with CCP such as fever and increase metabolic demand, severe anemia, shock, and sepsis [18]. Other common findings in the ECG in the CCP bitches was low voltage of the QRS complex and of P and T waves. Low-voltage ECG waves are potentially related to the presence of excess of fluid in the uterus, damping the electrical signal as described in the pleural effusion, peripheral edema, and/or ascites [16,29]. In human medicine, the decrease of sodium in sepsis patients can be an explanation for decreases in QRS voltage and increases in QRS duration because muscle cells become less excitable [30]. It is possible that even in bitches affected with pyometra there is a decrease in excitability of the myocardium due to the lowering of sodium levels. Bitches belonging to CCP showed blood sodium value within the reference ranges, although close to the minimum [21]. Ventricular premature complexes (VPCs) were recorded in bitches with CCP. VPCs are the most common arrhythmias in dogs [18]. VPCs originate from myocardium or specialized conduction tissue of the ventricular [18]. VPCs do not propagate through the normal conduction system; instead, the electrical impulses travel across working myocytes to depolarize both ventricles, which causes wide and differently shaped QRS complexes. In dogs, VPCs could be the combined result of multiple factors including myocyte oxygen deficiency, myocardial damage, and endotoxemia that can occur in CCP [18]. Increased T wave amplitude was observed in 17.9% of dogs with CCP, however, 5/7 also had hyperkalemia. T wave amplitude changes often occur in conditions of hyperkalemia (tended T wave), or in association with left ventricular enlargement or ectopic ventricular beats [18]. ST depression >0.20 mV was observed in 10.2% of dogs with CCP. Different factors may be cause of the ST tract depression in dogs, such as lack of oxygenation of the myocardium, pericarditis, myocarditis, hyperkalemia, ischemia (without necrosis), intraventricular conduction delay, and hypertrophy of the left ventricle [18]. Atrioventricular blockages were observed only in some bitches affected with CCP (3/39; 7.6%). The first degree of AV block is an increase in the PR interval, which represents a delay in the propagation of the impulse from the atria to the ventricles [18]. The second-degree atrioventricular block is an intermittent interruption of atrioventricular conduction. They are classified according to a P/QRS ratio in four types: Mobitz I (or Wenckebach), Mobitz II, 2:1 block, and advanced degree. In dogs with CCP, atrioventricular blockages suggest inflammatory or degenerative processes affecting the cardiac conduction tissue [18]. The bifida morphology and the increase in the duration of the P wave over 0.4 s suggests left atrial enlargement, which is shown by some bitches of the CCP group (3/39, 7.6%). Sinus bradycardia was demonstrated in 5.1% of bitches of the CCP group. This ECG finding can be caused by damage to the sinoatrial node, or it can be found in dogs that are dying. Atrial fibrillation occurred 5.12% bitches in the CCP group. It is a rapid and uncoordinated atrial activation that paralyzes the atrium itself [18]. Atrial fibrillation is characterized by irregular R-R intervals, the absence of P waves, and the presence of F waves. Electrophysiological causes are represented by an increase of triggered activity and activation of re-entry mechanisms [18]. Often this arrhythmia is found in cardiomyopathies involving fibrosis or atrial enlargement [18]. The increase in the QT interval was seen in 5.12% of dogs with pyometra, in the absence of tachycardia. The QT interval is the cardiac electrical systole, including both depolarization and ventricular repolarization. An increase in the QT interval in dogs can be due to hypocalcaemia or taking drugs that increase QT [18].

## 5. Conclusions

The results of this study confirm the hypothesis that canine closed cervix pyometra is associated with cardiac arrhythmias and abnormalities on ECG recording, suggestive of myocardial injury. Further studies are necessary to confirm the exact incidence of myocardial damage associated with pyometra. For definitive diagnosis, heart muscle biopsy is often required, but it is not an acceptable procedure due to its high risk and costs.

Further studies should be undertaken to elucidate these topics. Moreover, there was variability in the clinical signs that showed the dogs, and this could be also considered as a limitation of this study. Ideally, in further studies, dogs would be grouped in terms of similar clinical signs.

## Figures and Tables

**Figure 1 vetsci-07-00183-f001:**
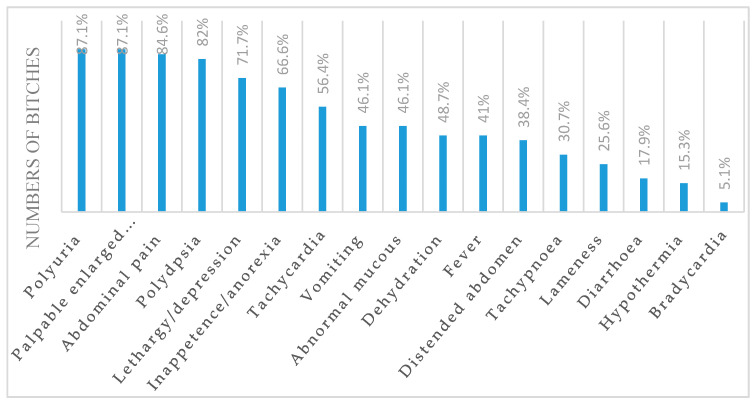
Clinical symptoms and signs in bitches affected with closed-cervix pyometra (CCP).

**Table 1 vetsci-07-00183-t001:** Mean and standard deviation of hematological variables analyzed for healthy bitches (CTR) and animals with closed cervix pyometra (CCP).

Hematological Parameters	CCP (No. 39)	CTR (No. 10)	Reference Ranges [20]
RBC (×10^6^/mm^3^)	3.40 ± 0.24 *	8.50 ± 0.73	5.5–8.5
Hemoglobin (g/dL)	7.2 ± 2.85 *	17.5 ± 1.23	12.0–18.0
PCV (%)	25.1 ± 6.44 *	49.3 ± 2.67	37.0–55.0
MCV (fl)	67.2 ± 7.2	64.6 ± 3.9	60.0–77.0
MCHC (g/dL)	31.7 ± 1.3 *	38.24 ± 1.87	32.0–36.0
RDW (%)	15.4 ± 2.5	12.10 ± 1.11	12.0–15.0
White blood cells (×10^3^/µL)	24.89 ± 3.45 *	11.9 ± 2.8	6.0–17.0
Neutrophils (×10^3^/µL)	12.10 ± 3.9 *	7.67 ± 1.49	3–10.0
Lymphocytes (×10^3^/µL)	3.7 ± 0.55	2.44 ± 0.56	1.0–4.8
Eosinophils (×10^3^/µL)	1.6 ± 0.14	1.3 ± 0.59	0.1–1.25
Monocytes (×10^3^/µL)	3.2 ± 0.03 *	0.71 ± 0.67	0.15–1.35
Platelet count (×10^3^/µL)	115 ± 23.65 *	289 ± 78.89	160–430

RBC = red blood cells, PCV = packed cell volume, MCV = mean corpuscular volume, MCHC = mean corpuscular hemoglobin concentration, RDW = red blood cell distribution width. * *p*-value ≤ 0.05.

**Table 2 vetsci-07-00183-t002:** Mean and standard deviation of biochemical variables analyzed for healthy bitches (CTR) and animals with closed cervix pyometra (CCP).

Biochemical Variables	CCP (No. 39)	CTR (No. 10)	Reference Ranges [21]
Albumin (g/dL)	2.40 ± 0.42 *	3.6 ± 0.34	2.6–3.3
Total protein (g/dL)	6.1 ± 1.78	6.2 ± 0.98	6.0–8.0
Globulin (g/dL)	3.5 ± 1.66	2.9 ± 2.1	2.7–4.4
Glucose (mmol/L)	4.9 ± 0.5	5.1 ± 0.6	4.5–5.8
ALT (IU/L)	98.2 ± 76.7 *	54.3 ± 12.1	21–73
ALP (IU/L)	102 ± 56 *	46.7 ± 18.1	20–156
GGT (IU/L)	2.8 ± 4.5	5.6 ± 1.11	1.2–6.4
Urea (mg/dL)	56.8 ± 24.1 *	25.3 ± 11.4	21.4–59.9
Creatinine (mg/dL)	1.8 ± 2.2 *	0.78 ± 0.32	0.5–1.5
Sodium (mmol/L)	145 ± 0.55 *	151 ± 1.2	145–153
Potassium (mmol/L)	4.2 ± 0.01 *	3.8 ± 0.02	3.7–4.1
Total Ca (mmol/L)	2.68 ± 0.22	2.80 ± 0.32	1.93 ± 3.03

ALT = alanine aminotrasferase, ALP = alkaline phosphatase, GGT = gamma-glutamyl transferase. * *p*-value ≤ 0.05.

**Table 3 vetsci-07-00183-t003:** Mean and standard deviation of electrocardiographic findings analyzed for both groups, and the results of the statistical test.

Electrocardiographic Variables	CCP (No. 39)	CTR (No. 10)	Reference Ranges [19]
Heart rate (bpm)	135.7 ± 37	86.2 ± 16.8	70–160
Mean electrical axis (degrees °)	71.4 ± 18.4	72.1 ± 14	40–100
P-wave duration (ms)	45 ± 0.75 **	40.8 ± 0.32 **	30–50
P-wave amplitude (mV)	0.2 ± 0.1 **	0.35 ± 0.12 **	<0.04
PR duration (ms)	96 ± 14 **	10.2 ± 3.18 **	60–130
QRS duration (ms)	55 ± 0.11	43.8 ± 4.6	Up to 0.5
R amplitude (mV)	0.78 ± 0.33 **	10.3 ± 2.42 **	0.5–2.5
ST deviation (mV)	0.00 ± 0.30	0	<0.2
T amplitude (mV)	0.16 ± 0.12	0.06 ± 0.05	<25% R Wave
QT duration (s)	220 ± 33	201 ± 7	150–250

** *p*-value ≤ 0.01.

**Table 4 vetsci-07-00183-t004:** Prevalence (in percentage) of ECG abnormalities in bitches suffering closed-cervix pyometra.

Electrocardiographic Parameters		CCP (%)
Heart rate (bpm)	High	56.4%
	Low	5.1%
P-wave duration (s)	High	5.1%
	Low	0%
P-wave amplitude (mV)	High	0%
	Low	43.5%
PR duration (sec)	High	7.6%
	Low	0%
QRS duration	High	23%
	Low	0%
QRS amplitude (mV)	High	5.1%
	Low	28.2%
ST deviation	Elevation	0%
	Depression	10.2%
T amplitude	High	17.9%
	Low	28.2%
QT duration	High	5.1%
	Low	0%

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
