# Peer review of "Electrocardiographic Findings in Bitches Affected by Closed Cervix Pyometra"

_vetsci, 2020, doi:10.3390/vetsci7040183_

Round 1

Reviewer 1 Report

The paper is for the Electrocardiographic findings in bitches affected by 3 closed-cervix pyometra. The paper is well organized and results sounds well.

Author Response

Dear Editor,

Dear Reviewer,

Thank you very much for your time and all your comments. We have revised the references and increased them. The English structure and grammar of the manuscript has been thoroughly reviewed.

We thank all of the reviewers for precise and thoughtful comments and constructive criticism, which has led to a better manuscript.

We revised the manuscript in relation to the suggestions of reviewers and more detailed answers are given below.

The changes made in the manuscript to address comments of all two reviewers are in red.

We do hope that the revised manuscript now suits for publication in Veterinary Sciences.

I’m looking forward to hearing from you.

Yours faithfully,

Annamaria Passantino on behalf of all the authors.

Reviewer 2 Report

The objective of this work was to report the electrocardiografic findings in bitches with closed-cervix pyometra (CCP). The purpose of this study was to describe electrocardiographic features in bitches affected with CCP and to assess the clinical relevance of ECG changes and to relate the severity of ECG with laboratory findings. The results obtained confirm the hypothesis that canine CCP is associated with cardiac arrhythmias and abnomalities on ECG results suggestive of miocardiac injury.

The paper needs carefull english edditing. Some exemples are marked in the text.

Only part  of the work is new (ECG alterations in pyometra cases) and a better effort should be put in the associations of clinical and laboratoty findings. Also justifications of ECG alterations should be improved and pratical implications should be point out, otherwise this work is mainly descriptive.

L46 – “Often anaesthesia and ovariohysterectomy are the only effective treatment of pyometra” – anaesthesia is not a treatment.  

L86 – “Haematological evaluations included haematocrit (HCT), red blood cell count (RBC), haemoglobin (HGB)…”  All these parameters are part of the hemograme. There is no need to point individually (we can see them in results).

M&M – Ecocardio shoud have been done to better caracterize possible heart alterations.

Figure 1 is not need as results are described in the test.

L132 – “All bitches were in the diestrus at the moment of diagnosis” – how was this confirmed? By histology?

L139 – The description of the inclusive criteria of SIRS should be writen in M&M.

L165 – “Regarding hepatic and renal function, high levels of ALT (P=0.038), ALP (P=0.041), urea (P=0.046), and creatinine (P=0.028) were detected in CPP in comparison with CTR”. Levels are not particular high, they are higher than ….

L167 – “Significant differences were also observed in the low level of sodium (P=0.05) and high-level potassium (P=0.05)”.  The diferences, if they exist, are not marked in table 2.

Figure 2 is not need as results are described in the test.

L197 – “It has been reported a total mortality of approximately 10% in bitches affected by pyometra [1]” – This is true but in cases when OVH was not performed.

L 202 to L 212 – Justification of haematological alterations are already well described in several articles. Maybe more evidence should be put in the association between the presence of SRIS (2,3 or 4 parameters included) and ECG alterations.

L216 – “Renal dysfunction is common in the CCP it shows polyuria and polydipsia associated with increased creatinine and urea”. This is not necessarily true. In fact most of the times there are no alterations in creatinina and urea (only if the reason is pre-renal).  Unfortunately, no bacteriological data is available. Polyuria and compensatory polydipsia are normal associated with the effect of the endotoxins in the kidney.

L229 -  “…the presence of myocardial induces…”. The presence of what?

L230 – “an increase of wall stress..” What is Wall stress?

L 238 – “..low voltage of the QRS 11/39 (28.2%) complex, P 17/39 239 (43.5%), and T 11/39 (28.2%) waves” . ..refrase

L234 – L 280:  Justifications of ECG alterations are too descriptive and not always associated with pyometra: exemple: “The decrease in the amplitude of the waves can occur due to the presence of pleural effusion, pneumothorax, pericardial effusion, the interposition of a significant amount of fat, decrease in myocardial mass (e.g. replacement of the myocardium with fibrous tissue)[16].”… “In humans, congenital causes of increase in QT are also recognized, as well as numerous other predisposing factors such as hypomagnesaemia, heart failure, old age and HIV infection [28].

L244 – “It is possible that even in bitches affected with pyometra there is a decrease in excitability of the myocardium due to the lowering of sodium levels” – however, Na levels are in the normal range …

L282 -  Authors state in conclusion that: “This is the first study that documents ECG recording, laboratory, and clinical findings in dogs with CCP”. This maybe true to ECG data, but clinical and laboratory findings are already well described. Please refrase.

Author Response

Dear Editor,

Thank you very much for your time and all your comments. We have revised the references and increased them. The English structure and grammar of the manuscript has been thoroughly reviewed.

We thank all of the reviewers for precise and thoughtful comments and constructive criticism, which has led to a better manuscript.

We revised the manuscript in relation to the suggestions of reviewers and more detailed answers are given below.

The changes made in the manuscript to address comments of all two reviewers are in red.

We do hope that the revised manuscript now suits for publication in Veterinary Sciences.

Response to Reviewer 2 comments

The objective of this work was to report the electrocardiografic findings in bitches with closed-cervix pyometra (CCP). The purpose of this study was to describe electrocardiographic features in bitches affected with CCP and to assess the clinical relevance of ECG changes and to relate the severity of ECG with laboratory findings. The results obtained confirm the hypothesis that canine CCP is associated with cardiac arrhythmias and abnomalities on ECG results suggestive of miocardiac injury.

The paper needs carefull english edditing. Some exemples are marked in the text.

Only part of the work is new (ECG alterations in pyometra cases) and a better effort should be put in the associations of clinical and laboratoty findings. Also justifications of ECG alterations should be improved and pratical implications should be point out, otherwise this work is mainly descriptive.

We would like to thank deeply the reviewer for his/her time and effort. We have accepted the suggestions and revised it. The English structure and grammar of the manuscript has been thoroughly reviewed. We hope that the current version of the manuscript has been improved.

L46 – “Often anaesthesia and ovariohysterectomy are the only effective treatment of pyometra” – anaesthesia is not a treatment.  

We have corrected it. The word “anaesthesia” has been delayed (L44).

L86 – “Haematological evaluations included haematocrit (HCT), red blood cell count (RBC), haemoglobin (HGB)…”  All these parameters are part of the hemograme. There is no need to point individually (we can see them in results).

We have corrected the text. All parameters of the hemogram have been delayed.

M&M – Ecocardio shoud have been done to better caracterize possible heart alterations.

In really, echocardiographic examination was performed in all dogs in order to exclude cardiovascular abnormalities. We have inserted it in the text (L82-83).

Figure 1 is not need as results are described in the test.

We have deleted clinical findings from text, as you have suggested.

L132 – “All bitches were in the diestrus at the moment of diagnosis” – how was this confirmed? By histology?

Of course, we have confirmed the diagnosis by histological examination (L145).

L139 – The description of the inclusive criteria of SIRS should be writen in M&M.

We have inserted the criteria in materials and methods (L88-92).

L165 – “Regarding hepatic and renal function, high levels of ALT (P=0.038), ALP (P=0.041), urea (P=0.046), and creatinine (P=0.028) were detected in CPP in comparison with CTR”. Levels are not particular high, they are higher than ….

We have modified the sentence (L172-175).

L167 – “Significant differences were also observed in the low level of sodium (P=0.05) and high-level potassium (P=0.05)”.  The diferences, if they exist, are not marked in table 2.

We have revised the table 2.

Figure 2 is not need as results are described in the test.

We have deleted the figure 2

L197 – “It has been reported a total mortality of approximately 10% in bitches affected by pyometra [1]” – This is true but in cases when OVH was not performed.

We have modified the sentence (L204-205).

L 202 to L 212 – Justification of haematological alterations are already well described in several articles. Maybe more evidence should be put in the association between the presence of SRIS (2,3 or 4 parameters included) and ECG alterations.

Other information has been added highlighting the existent correlation between SIRS and ECG (L219-222).

L216 – “Renal dysfunction is common in the CCP it shows polyuria and polydipsia associated with increased creatinine and urea”. This is not necessarily true. In fact most of the times there are no alterations in creatinina and urea (only if the reason is pre-renal).  Unfortunately, no bacteriological data is available. Polyuria and compensatory polydipsia are normal associated with the effect of the endotoxins in the kidney.

We have modified the sentence (L225-227).

L229 -  “…the presence of myocardial induces…”. The presence of what?

We have added injury after myocardial (L239).

L230 – “an increase of wall stress..” What is Wall stress?

Wall stress represents the systolic force or work per surface unit. It is the systolic force made by myocardial tissues. Therefore, stress increase indicates enlargement of the left ventricle or increase of intracavitary pressure.

We hope to have clarified the concept. In the text we have added that wall stress is referred to myocardium (L240).

L 238 – “..low voltage of the QRS 11/39 (28.2%) complex, P 17/39 239 (43.5%), and T 11/39 (28.2%) waves” . ..refrase

The sentence has been rephrased (L248).

L234 – L 280:  Justifications of ECG alterations are too descriptive and not always associated with pyometra: exemple: “The decrease in the amplitude of the waves can occur due to the presence of pleural effusion, pneumothorax, pericardial effusion, the interposition of a significant amount of fat, decrease in myocardial mass (e.g. replacement of the myocardium with fibrous tissue)[16].”… “In humans, congenital causes of increase in QT are also recognized, as well as numerous other predisposing factors such as hypomagnesaemia, heart failure, old age and HIV infection [28].

Some sentences have been deleted and others modified (L249-286).

L244 – “It is possible that even in bitches affected with pyometra there is a decrease in excitability of the myocardium due to the lowering of sodium levels” – however, Na levels are in the normal range …

The sentence has been modified (L252-255).

L282 -  Authors state in conclusion that: “This is the first study that documents ECG recording, laboratory, and clinical findings in dogs with CCP”. This maybe true to ECG data, but clinical and laboratory findings are already well described. Please refrase.

The sentence has been rephrased (L289-291).

Reviewer 3 Report

This paper examines the electrocardiographic patterns in bitches of various breeds suffering closed-cervix pyometra and its association with serum metabolites and previous clinical findings. This is an excellent effort by the authors to understand the impacts of closed-cervix pyometra on the hearth functioning and metabolite responses in bitches, and provides some new knowledge that will help to develop a suitable strategy for using electrocardiography in canine obstetrics to identify pathological outcomes before they become fatal.

Having said that, the authors have appropriately articulated the research objectives and have conducted a well-designed investigation leading to important findings. However, The manuscript does not read well in general and is not written meticulously with a lack of details on  materials and methods (e.g. management of the electrocardiogram, statistical analyses, etc). There are a few minor changes that need to be incorporated and may be helpful for the readers.

Introduction: The authors need to better articulate and provide rationale why it is important to deepen our knowledge on the canine electrocardiographic changes as well as some serum metabolites in the course of close-cervix pyometra and healthy bitches. Materials and Methods:

There is not enough detail on the electrocardiographic sampling (how were electrodes attached to the skin, solution used to attain proper contact of the electrodes to magnify electrical transmission, hours of  examinations, type of calibrated paper, etc.). Details on the equipment used for blood analyses are required. Kits and procedures for determination of serum metabolites, minerals, and enzymes need to be described in detail.

Statistical analysis: Data normality was assessed but nothing is said about the normality of the data analyzed. Authors must indicate if they used a completely randomized design or at-student test for continuous variables. They had the bodyweight of bitches. Could this variable be used as a covariate? Additionally, given the multiple serum variables, a multivariate analysis e.g. PCA would be beneficial to show how the groups differ in blood chemistry in a multivariate space and would add a nice component to the paper.

Discussion: Many values already presented in results are repeated in the discussion section; therefore, the discussion needs some improvements. It should be clearly written why did the authors chose so many blood variables for this study – is there a scientific reason to examine so many blood variables? The variability of blood metabolites among groups needs to be discussed in more detail, explaining the reasons for these variabilities. The discussion is largely speculative and the text requires linguistic check.

Additional observations are indicated in the text or listed below:

Indicate if bitches had previous normal pregnancies and delivered healthy puppies at term.

Indicate how many consecutive beats were measured with the electrocardiogram.

Indicate statistical procedures for analyses of continuous variables.

Include a vertical line for the Y-axis in Figure 1.

Eliminate values above bars. Include instead percentages, e.g. 87.1% for first bar.

Include a vertical line for the Y-axis in Figure 2. Place percentages above bars.

What is your point regarding “CCP showed blood sodium value within the reference ranges although close to the minimum”

Author Response

Dear Editor,

Thank you very much for your time and all your comments. We have revised the references and increased them. The English structure and grammar of the manuscript has been thoroughly reviewed.

We thank all of the reviewers for precise and thoughtful comments and constructive criticism, which has led to a better manuscript.

We revised the manuscript in relation to the suggestions of reviewers and more detailed answers are given below.

The changes made in the manuscript to address comments of all two reviewers are in red.

We do hope that the revised manuscript now suits for publication in Veterinary Sciences.

Response to Reviewer 3 comments

This paper examines the electrocardiographic patterns in bitches of various breeds suffering closed-cervix pyometra and its association with serum metabolites and previous clinical findings. This is an excellent effort by the authors to understand the impacts of closed-cervix pyometra on the hearth functioning and metabolite responses in bitches, and provides some new knowledge that will help to develop a suitable strategy for using electrocardiography in canine obstetrics to identify pathological outcomes before they become fatal.

Having said that, the authors have appropriately articulated the research objectives and have conducted a well-designed investigation leading to important findings. However, The manuscript does not read well in general and is not written meticulously with a lack of details on  materials and methods (e.g. management of the electrocardiogram, statistical analyses, etc). There are a few minor changes that need to be incorporated and may be helpful for the readers.

We would like to thank the reviewer for the positive evaluation of the work and for encouragements. We highly appreciate it, as well as we do find the comments very useful to improve the paper.

Introduction: The authors need to better articulate and provide rationale why it is important to deepen our knowledge on the canine electrocardiographic changes as well as some serum metabolites in the course of close-cervix pyometra and healthy bitches.

The introduction has been better articulate providing the rationale on importance of ECG in close-cervix pyometra (L48-53).

Materials and Methods:

There is not enough detail on the electrocardiographic sampling (how were electrodes attached to the skin, solution used to attain proper contact of the electrodes to magnify electrical transmission, hours of examinations, type of calibrated paper, etc.). Details on the equipment used for blood analyses are required. Kits and procedures for determination of serum metabolites, minerals, and enzymes need to be described in detail.

Details on the electrocardiographic sampling have been added (L108-111). Details, kits and procedures have been included (L99-101; 102-103).

Statistical analysis: Data normality was assessed but nothing is said about the normality of the data analyzed. Authors must indicate if they used a completely randomized design or at-student test for continuous variables. They had the bodyweight of bitches. Could this variable be used as a covariate? Additionally, given the multiple serum variables, a multivariate analysis e.g. PCA would be beneficial to show how the groups differ in blood chemistry in a multivariate space and would add a nice component to the paper.

We have revised this part of the text and added data distribution (L136). In relation to the bodyweight of bitches, in our opinion it has not been appropriate to consider the body weight as covariate because it may be hardly influence the onset of ECG alteration. Indeed, no correlation statistically significant has been observed.

In addition, in regard to your suggestion on using of a multivariate analysis in order to highlight difference between the several serum variables, it is well accepted for future investigations.

Discussion: Many values already presented in results are repeated in the discussion section; therefore, the discussion needs some improvements. It should be clearly written why did the authors chose so many blood variables for this study – is there a scientific reason to examine so many blood variables? The variability of blood metabolites among groups needs to be discussed in more detail, explaining the reasons for these variabilities. The discussion is largely speculative and the text requires linguistic check.

The discussions have been revised including the English language. Considered blood variables are made as pre-surgery check-up.

Additional observations are indicated in the text or listed below:

Indicate if bitches had previous normal pregnancies and delivered healthy puppies at term.

We have indicated it (L71-72).

Indicate how many consecutive beats were measured with the electrocardiogram.

We have indicated it (L116-117).

Indicate statistical procedures for analyses of continuous variables.

There was a typo. We have modified it (L133).

Include a vertical line for the Y-axis in Figure 1.

Figure 1 has been amended according to the other Reviewer’s suggestion.

Eliminate values above bars. Include instead percentages, e.g. 87.1% for first bar.

We have modified it (L142, See figure 1).

Include a vertical line for the Y-axis in Figure 2. Place percentages above bars.

Figure 2 has been deleted according to the Reviewer 2.

What is your point regarding “CCP showed blood sodium value within the reference ranges although close to the minimum”.

The sentence has been clarified (L254-255).

I’m looking forward to hearing from you.

Yours faithfully,

Annamaria Passantino on behalf of all the authors.

Round 2

Reviewer 2 Report

Authors have made alterations according with the reviewers sugestions. The text was edited.

The article is ready to be accepted

Reviewer 3 Report

The authors have substantially improved their manuscript following the suggestion of this reviewer. Therefore, This reviewer considers that this manuscript is apt to be published in Veterinary Sciences in its present form.